# Young people's preferences for the use of emerging technologies for asymptomatic regular chlamydia testing and management: a discrete choice experiment in England

Sue Eaton,[1] Deborah Biggerstaff,[1] Stavros Petrou,[1] Leeza Osipenko,[2] Jo Gibbs,[3] Claudia S Estcourt,[4] Tariq Sadiq,[5] Ala Szczepura[6]

For numbered affiliations see end of article.

**Correspondence to**
Professor Ala Szczepura;
ala.szczepura@coventry.ac.uk

## ABSTRACT

**Objective** To undertake a comprehensive assessment of the strength of preferences among young people for attributes of emerging technologies for testing and treatment of asymptomatic chlamydia.

**Design** Discrete choice experiment (DCE) with sequential mixed methods design. A staged approach to selection of attributes/levels included two literature reviews, focus groups with young people aged 16–24 years (n=21), experts' review (n=13) and narrative synthesis. Cognitive testing was undertaken to pilot and adapt the initial questionnaire. Online national panel was used for final DCE survey to maximise generalisability. Analysis of questionnaire responses used multinomial logit models and included validity checks.

**Setting** England.

**Participants** 1230 young people aged 16–24 from a national online panel (completion rate 73%).

**Outcome measures** ORs for service attributes in relation to reference levels.

**Results** The strongest attribute influencing preferences was chlamydia test accuracy (OR 3.24, 95% CI 3.13 to 3.36), followed by time to result (OR 1.81, 95% CI 1.71 to 1.91). Respondents showed a preference for remote chlamydia testing options (self-testing, self-sampling and postal testing) over attendance at a testing location. For accessing treatment following a positive test result, there was a general preference for online (OR 1.21, 95% CI 1.15 to 1.28) versus traditional general practitioner (OR 1.18, 95% CI 1.12 to 1.24) or pharmacy (OR 1.15, 95% CI 1.10 to 1.22) over clinic services. For accessing a healthcare professional and receipt of antibiotics, there was little difference in preferences between options.

**Conclusions** Both test accuracy and very short intervals between testing and results were important factors for young people when deciding whether to undergo a routine test for asymptomatic chlamydia, with test accuracy being more important. These findings should assist technology developers, policymakers, commissioners and service providers to optimise technology adoption in service redesign, although use of an online panel may limit generalisability of findings to other populations.

### Strengths and limitations of this study

► To our knowledge, this is the first large-scale discrete choice experiment (DCE) study to examine the preferences of a geographically representative national sample of young people for emerging technologies designed to improve screening for and treatment of asymptomatic chlamydia.

► Robust sequential methods were used to select final DCE attributes and levels, including two literature reviews, focus groups with young people and review by expert groups.

► An online panel enabled access to the population targeted for screening, including young people with no personal experience of chlamydia testing and treatment.

► A limitation of this work is that the literature reviews inevitably identified more potential attributes than could be included in the DCE.

► The use of an online panel may limit generalisability of findings to the very small percentage (3%) of 15–24 year-olds who do not currently access the internet, and therefore over-represent the acceptability of online care.

## INTRODUCTION

Chlamydia is the most commonly reported sexually transmitted infection (STI) in England, with young people aged 15–24 accounting for 63% of diagnoses in 2016.[1] The estimated annual cost of chlamydia treatment in 2015 was £249.8 million.[2] Undetected infections and reinfections can lead to significant adverse health consequences, such as pelvic inflammatory disease, ectopic pregnancy and infertility, which have an impact on National Health Service (NHS) costs and health-related quality of life.[3] A National Chlamydia Screening Programme (NCSP) was therefore rolled out in England for 16–24 year-olds between 2003 and 2008.[4]

However, despite recent evidence of a reduction in chlamydia prevalence in England for 2000–2015, concurrent with large-scale population testing,[5] and the fact that uncomplicated infection is easy to treat with oral antibiotics, uptake of screening for chlamydia remains low.[6] Worryingly, there has also been a decline over the last 4 years in the number of local authorities achieving the public health outcomes framework indicator of 2300 diagnoses per 100 000 population.[1] Barriers identified for young people accessing STI testing and treatment services include tangible service attributes, such as location of service, and personal or behavioural factors such as the stigma associated with attendance, embarrassment, fear of being recognised and privacy concerns.[7–9]

A range of options are currently available for young people to access asymptomatic chlamydia testing and treatment services. For testing, options include attending a genitourinary medicine (GUM) clinic, testing via primary care, for example, pharmacies/general practitioners (GP), or via internet testing services where a number of websites offer free self-sampling kits online with samples sent to laboratories for analysis and results communicated directly to patients. For individuals who screen positive, options for accessing antibiotic treatment include attending a GUM/sexual health clinic, GP practice or pharmacy. Even though a national screening programme exists in England, large geographical variations exist in testing coverage, from 16% of young people tested for chlamydia in the West Midlands region to 27% in London, and in the proportion of positive cases treated.[1 10]

For many health services, digital technology is now widely regarded by policymakers as one approach to improve access and reduce costs.[11] For chlamydia testing and treatment, a range of digital options are available, and more technologies are likely to enter the market within the next 3–5 years.[12] These include point-of-care tests,[13] online postal self-sampling, eSexual health clinics,[14] apps and non-face-to-face consultation methods[15 16] and self-tests networked through mobile phones.[17] Such innovations provide an opportunity to redesign current chlamydia screening services with the aim of improving testing and treatment uptake. However, such service redesign should be based on a sound understanding of the preferences of young people as service users for specific attributes of such services.

Research specifically measuring young people's preferences nationally for chlamydia testing and treatment service options is lacking. Even for STI testing services generally, relatively few studies have used a discrete choice experiment (DCE) design to assess preferences in the UK population.[18–21] In comparison with other preference elicitation methods, a DCE can quantify the relative importance of different attributes that characterise a new or existing product and/or service, identifying which attributes people prioritise and which they may be willing to trade with the view to maximising their utility.[22] A DCE requires respondents to choose between competing scenarios, for example, service options, described in terms of a particular attribute (eg, time to test result) and a range of levels (eg, 30 min to 14 days) and to compare these against an alternative scenario. DCE studies are very useful because they allow a direct assessment of relative preferences for various existing and hypothetical new service configurations or treatment approaches.[23]

The aim of the present study was to undertake a comprehensive assessment of the preferences of young people, targeted by the NCSP, for emerging technology options for testing and treatment in the context of a 'check-up test' where remote care could be medically appropriate.[14] To our knowledge, no previous study has attempted to disentangle strength of preference for attributes associated with new and emerging options for chlamydia screening (such as self-testing) or treatment (such as online prescription). Because STI services are 'open access' and not subject to gatekeeping by referral from a clinician, the impact of such disruptive innovations will be directly dependent on population preferences. Early insight into attributes that could influence whether individuals are more likely to use a new testing or treatment pathway should therefore be helpful in informing product development and pathway redesign for future chlamydia screening service models.[24 25]

## METHODS

This study was conducted using an exploratory sequential mixed methods design, adopting recommended stages for undertaking a DCE,[26] as shown in figure 1. In selecting methods to design the questionnaire and conduct the DCE, reference was made to the International Society for Pharmacoeconomics and Outcomes Research good practice checklist for conjoint analysis.[27]

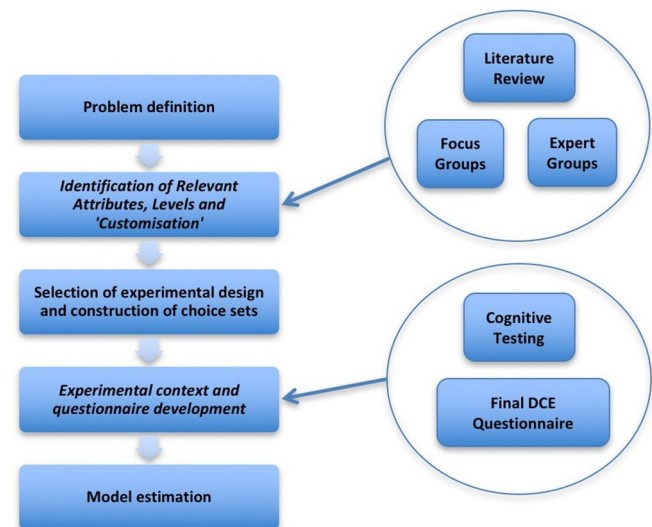

**Figure 1** Stages in conducting a discrete choice experiment (DCE).

### Patient and public involvement

Young people (with and without experience of STI services) participated in focus groups to inform priorities included in the DCE; in cognitive testing to finalise questionnaire design; and in questionnaire completion to identify preferences, as described in the Acknowledgements section.

### Selection of attributes and levels

The attributes and associated levels were determined using a four-stage approach.[28] More detailed information is provided in online supplementary file 1.

### Literature reviews

Two literature reviews were undertaken to produce a comprehensive list of potential themes and factors that might influence young people's choices. These included: (1) a systematic review of the use of stated preference studies for STI testing and treatment (PROSPERO Reg: CRD42014014862); and (2) a scoping review of other research exploring preferences and acceptability of STI testing and treatment services.

### Focus groups

Focus groups were run with young people aged 16–24 years (4 groups, 3–7 per group; total n=21) to identify which themes and factors young people consider important when choosing to test for STIs. Convenience sampling was used to identify participants. Focus group topic guides incorporated typical vignettes of situations individuals might encounter. Sessions were recorded, transcribed and thematic analysis was performed to produce a list of potential attributes and levels.[29]

### Expert groups

Four expert groups were convened (n=13), including a range of professionals with expert knowledge of the service and technology context. Expert groups were asked to review the focus group findings in terms of clinical feasibility and practicality.[26]

### Narrative synthesis

Narrative synthesis[30] enabled outputs from the three prior stages to be synthesised for each potential attribute. This approach was adopted as it offered a clear approach to synthesis based on the following stages: (1) identification of a checklist of properties against which attributes can be considered; (2) tabulation against the checklist; and (3) conceptual mapping and triangulation against the checklist. The final synthesis highlighted a tension between young people's desire to be tested for a wide range of STIs irrespective of risk, versus clinical guidelines for selective testing of STIs based on population group prevalence and risk. The range of STIs presented in the DCE was consequently limited to chlamydia. Because focus group findings indicated difficulty in understanding several dimensions of test performance, test accuracy was expressed in terms of the likelihood of a false negative result.

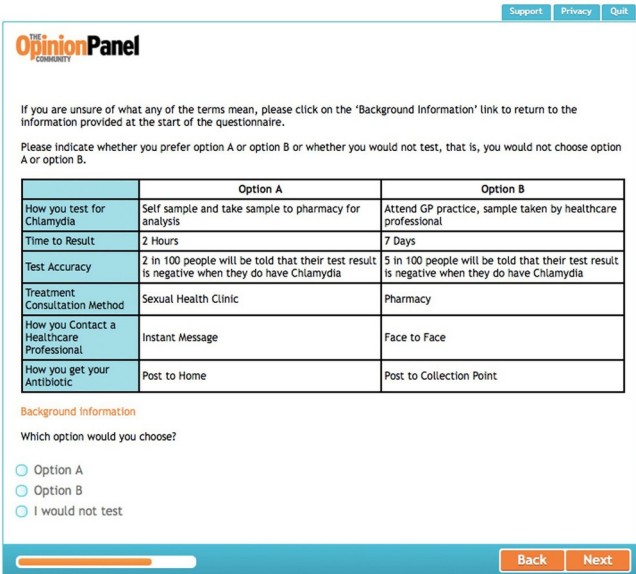

**Figure 2** Sample choice set from the discrete choice experiment.

### Questionnaire design and piloting

A generic pairwise choice with opt-out question was selected for the questionnaire design. Respondents were presented with a series of choice sets for which there were three responses: 'option A', 'option B' or 'I would not test'. A sample choice set is illustrated in figure 2. The questionnaire adopted a main effects design using full profiles (all attributes included in the study). While some DCEs do include an attribute on economic costs, DCEs exploring preferences for STI testing services in England have generally excluded cost since the NHS provides treatment 'free at the point of delivery.'[20 21] A cost attribute was therefore not included.

The questionnaire was developed using SAS V.9.4 software to ensure that the design was D-efficient.[27] A full factorial design was ruled out in favour of a fractional factorial design because a full factorial design would have contained 3072 possible alternatives, which would have been unmanageable in practice for individuals to complete or for a blocked questionnaire format to handle.[31] The smallest 100% efficient design that could be created included 48 choice sets. These were blocked (halved) into two questionnaires each with 24 choice sets using SAS JMP Pro V.9.2.0. Choice set 1 was repeated as choice set 25 in each questionnaire to provide an internal validity check.

Three rounds of cognitive testing (n=9) were undertaken to check respondents' comprehension of information when making choices. Cognitive testing, undertaken based on two questionnaires of 24 sets, confirmed that a study based on two such questionnaires was acceptable to participants. Some modifications were made to levels where reasons for choice selection demonstrated that one level (eg, 8 in 100 false negatives) dominated the reason for selection. Cognitive testing also identified that implausible combinations (eg, a postal test providing

**Table 1** Summary of attributes and levels included in the study

| Attribute | A1. How you test | A2. Time to result | A3. Accuracy | A4. Consultation method | A5. Access to HCP | A6. How you get antibiotics |
|---|---|---|---|---|---|---|
| L evel | | | | | | |
| L1 | Self-test | 30 min | 2 in 100 (false negative) | Online consultation | Phone | Post to home |
| L2 | Self-sample and post | 2 hours | 5 in 100 (false negative) | Pharmacy | IM | Post to collection point |
| L3 | Self-sample and pharmacy | 7 days | | GP | Email | Collect from pharmacy |
| L4 | Self-sample and education/work | 14 days | | Sexual health clinic | Face to face | Collect from sexual health clinic |
| L5 | GP practice—sample taken by healthcare professional | | | | | |
| L6 | Sexual health clinic—sample taken by healthcare professional | | | | | |

Full descriptions provided in online supplementary file 2 (pp 3–7).
GP, general practitioner; HCP, healthcare professional; IM, instant messenger.

a result in 2 hours) impacted on completion of the task, while unlikely (but feasible) combinations did not. Only implausible combinations were therefore excluded from the choice sets. The D-efficiency of the design[27] calculated by SAS JMP Pro V.9.2.0, prior to the removal of implausible combinations, was 98.06. The final combination selected for the questionnaire was the one which created no duplicate choices and which provided an equal balance of the number of choice sets containing overlap between questionnaires.

Respondents were asked to provide sociodemographic information, including their age, gender, ethnicity, region of residence, sexual preference, relationship status and whether they had previously been tested for an STI. An introduction to the DCE questionnaire provided background information and an explanation of the attributes and levels. The introduction was included in the cognitive testing rounds to check comprehension within the target age range. As a result, the text was modified and diagrams added to illustrate the chlamydia testing pathways. The final phase of testing demonstrated that the number of choice sets was acceptable to respondents. The use of an online panel also provided completion time data to support the internal validity checks, and enabled an accurate record of time taken to complete the survey.

### Final attributes and levels selected
The six attributes and 24 levels selected for inclusion in the final DCE questionnaire are shown in table 1. More detailed information used to explain these attributes and levels is provided in online supplementary file 2 (pp 3–7).

### Participants and recruitment procedure
Participants were drawn from the general population rather than from healthcare settings, thereby providing access to young people aged 16–24 years who had not previously been tested for an STI. An online national panel (YouthSight)[32] was used to maximise geographical reach and generalisability. Ex ante subgroup analysis

was planned by three age bands (16–18, 19–21, 22–24) and by gender (male, female). Where subgroup analysis is planned, it is recommended that there should be a minimum of 200 respondents in each subgroup, so a sample size of 1200 was required in our study, 600 per questionnaire.[33] Consent was obtained online prior to questionnaire completion. Participants were offered a small reimbursement of one point (equivalent to £1) for completion of a survey of up to 20 min in length. Points could be exchanged for shopping vouchers.

### Statistical analyses
The multinomial logit (MNL) model developed by McFadden was used for analysing responses; this is recognised as the convention for three-response choice set studies ('option A', 'option B' or 'I would not test').[26 34] Analysis used STATA V.13 SE with the method and code outlined by Ryan et al.[23] Analyses presented ORs, 95% CIs and coefficients for each attribute level. Variables within the MNL model were all treated as categorical variables for the analysis. Attribute levels were specified using dummy coding, the preferred form of coding where ORs are to be calculated. Within the model, the levels that were dropped to form the reference levels reflected those aligned to a 'typical' sexual health clinic pathway (summarised in table 2). To test the internal validity of questionnaire responses, analyses compared full results against: (1) removal of responses where participants did not answer the repeated choice set consistently; (2) removal of any respondents who took less than the minimum time (5 min) observed in cognitive testing to complete the questionnaire; and (3) removal of responses containing the opt-out question data. Further tests for internal consistency and rationality were not included, since excluding responses on this basis may be viewed as an inappropriate imposition of rationality.[35]

Demographic characteristics (gender, age and ethnicity) of respondents were compared with national census

**Table 2** Dummy coding reference level

| Attribute | Reference level |
|---|---|
| How you test | Sexual health clinic |
| Time to result | 7 days |
| Accuracy | 5 in 100 (false negative) |
| Consultation method | Sexual health clinic |
| Access to HCP | Face to face |
| How you get antibiotics | Collect from sexual health clinic |

HCP, healthcare professional.

data.[36] The influence of patient-level characteristics (age, sex and STI testing history (yes/no)) on the likelihood of not choosing to test was examined. In addition to *ex ante* planned subgroup analyses, if sufficient responses were received analysis was also planned to compare: (1) respondents who had, or had not, previously tested for an STI, and (2) those who indicated their relationship status as 'single' versus those in a sexual relationship with one person.

Trade-off between accuracy and time to result was examined by considering the probability of uptake for tests with characteristics at the opposite ends of the spectrum, that is, 'lower accuracy (5% false negatives), faster time to result (30 min)' and 'higher accuracy (2% false negatives), longer time to result (14 days)'.

## RESULTS

In total, 1230 fully completed questionnaires were received, the platform analytics showed that 460 people had opened the questionnaire but did not complete it, providing a completion rate of 73%. No further information was available on the demographics of the 460 non-responders nor any information on the point at which they chose to exit the survey, so these people could not be included in our analyses. Time to complete the 25 choice sets ranged from 1 min 19 s to 30 min 19 s with a median time to completion of 7 min 51 s. Demographic characteristics of respondents are summarised in the online supplementary file 3 (part 1). Comparing the DCE sample to the characteristics of 16–24 year-olds in England indicates that gender, age and ethnicity are broadly in line with national population demographics, with a slight under-representation of the 16–18 years age group. In terms of the geographical residence of participants, the sample was broadly in line with the geographical distribution of 16–24 year-olds identified in the Office for National Statistics Mid-Year Population Estimates 2015,[37] with the exception of a lower proportion of respondents from the East of England.

Internal validity checks showed that, in comparison to the full data set, all checks yielded very similar ORs with no change in the order of the strength of preference for a level within each attribute. Results of the internal validity checks are included in the online supplementary file 3 (part 2).

OR values for the full data set and subgroups analysed are presented in table 3 with reference levels (1.00). Analysis of the trade-off between accuracy and time to result is presented in table 4. More detailed DCE results, including 95% CIs as well as OR values plus coefficients for each attribute level for all subgroup analyses, are presented in online supplementary file 3 (part 3).

In table 3, looking across all attributes and levels for the whole population, participants expressed the strongest preference for attribute 1—accuracy of test result (OR 3.242, 95% CI 3.130 to 3.359). There are also differences in the strength of preference between males and females for this attribute (OR 2.951, 95% CI 2.807 to 3.101 and OR 3.570, 95% CI 3.396 to 3.753, respectively), and between those who had previously tested or not tested (OR 3.000, 95% CI 2.820 to 3.191 and OR 3.482, 95% CI 3.331 to 3.640, respectively). Time to result was the attribute showing the next strongest preference across all subgroups (OR 1.806, 95% CI 1.711 to 1.906). These results are consistent with the logical expectation that people will prefer higher accuracy and a shorter waiting time. Looking specifically at the trade-off between accuracy and time to result, table 4 indicates that participants are willing to wait noticeably longer in order to have a test result with a lower chance of a false negative result.

When considering how to test, all subgroups demonstrated a preference for self-testing (OR 1.618, 95% CI 1.514 to 1.729) over attendance at a sexual health clinic. Testing via an outreach service in an educational/work setting was found to be the least preferred option (OR 0.821, 95% CI 0.773 to 0.872). Respondents showed a consistent strength of preference for those options that do not involve direct contact with healthcare professionals (self-testing and postal self-sampling), compared with those that do. For consultation and treatment following a positive test result, there was a preference for non-sexual health clinic pathways with online consultation to access treatment (OR 1.212, 95% CI 1.150 to 1.277), treatment via general practice (OR 1.183, 95% CI 1.123 to 1.246) and treatment via pharmacy (OR 1.158, 95% CI 1.100 to 1.220) preferred in the full data set. At subgroup level (age, gender, testing history and relationship status), more variation was found in the preference order for this attribute, with the exception of the sexual health clinic which was consistently the least preferred option across all subgroups (see table 3 and online supplementary file 3, part 3). Finally, the full data set shows that, if someone wants to access a healthcare professional, there was no statistically significant preference for instant messenger or email access compared with accessing the professional face to face. Telephone access to a healthcare professional was the least preferred access option (OR 0.949, 95% CI 0.903 to 0.998), and the only statistically significant result when compared with face-to-face access (the reference level). There was, similarly, no statistically significant difference in preferences for how young people might

**Table 3** Overview ORs for all subgroups included within the analysis

| Attribute and levels | Full data set | Male | Female | 16–18 | 19–21 | 22–24 | Single | One partner | Previous test | No previous test |
|---|---|---|---|---|---|---|---|---|---|---|
| | 1230 | 607 | 623 | 415 | 406 | 409 | 615 | 512 | 393 | 790 |
| | OR | OR | OR | OR | OR | OR | OR | OR | OR | OR |
| **1. How you test** | | | | | | | | | | |
| Self-test | 1.618 | 1.549 | 1.693 | 1.664 | 1.346 | 1.895 | 1.632 | 1.712 | 1.524 | 1.678 |
| Self-sample and post to a laboratory for analysis | 1.358 | 1.308 | 1.411 | 1.477 | 1.138 | 1.487 | 1.380 | 1.380 | 1.178 | 1.465 |
| Self-sample and take to a pharmacy for analysis | 1.155 | 1.131 | 1.180 | 1.270 | 1.021 | 1.185 | 1.220 | 1.109 | 0.945 | 1.277 |
| Self-sample and take to place of education/work for analysis | 0.821 | 0.919 | 0.733 | 0.804 | 0.806 | 0.852 | 0.851 | 0.778 | 0.761 | 0.861 |
| GP practice—sample taken by a healthcare professional | 1.019 | 1.049 | 0.990 | 1.085 | 1.035 | 0.941 | 0.994 | 1.066 | 0.885 | 1.101 |
| Sexual health clinic—sample taken by a healthcare professional | 1.000 | 1.000 | 1.000 | 1.000 | 1.000 | 1.000 | 1.000 | 1.000 | 1.000 | 1.000 |
| **2. Time to result** | | | | | | | | | | |
| 30 min | 1.806 | 1.828 | 1.786 | 1.763 | 1.888 | 1.772 | 1.730 | 1.896 | 1.808 | 1.797 |
| 2 hours | 1.402 | 1.365 | 1.442 | 1.326 | 1.548 | 1.344 | 1.311 | 1.516 | 1.357 | 1.423 |
| 7 days | 1.000 | 1.000 | 1.000 | 1.000 | 1.000 | 1.000 | 1.000 | 1.000 | 1.000 | 1.000 |
| 14 days | 0.862 | 0.844 | 0.881 | 0.878 | 0.882 | 0.828 | 0.862 | 0.862 | 0.882 | 0.856 |
| **3. Accuracy** | | | | | | | | | | |
| 2 in 100 false negatives | 3.242 | 2.951 | 3.570 | 3.307 | 3.161 | 3.282 | 3.275 | 3.305 | 3.000 | 3.482 |
| 5 in 100 false negatives | 1.000 | 1.000 | 1.000 | 1.000 | 1.000 | 1.000 | 1.000 | 1.000 | 1.000 | 1.000 |
| **4. Consultation method** | | | | | | | | | | |
| Online consultation | 1.212 | 1.187 | 1.239 | 1.187 | 1.200 | 1.253 | 1.156 | 1.285 | 1.174 | 1.228 |
| Pharmacy consultation | 1.158 | 1.106 | 1.215 | 1.167 | 1.209 | 1.102 | 1.115 | 1.225 | 1.098 | 1.183 |
| GP consultation | 1.183 | 1.172 | 1.194 | 1.279 | 1.142 | 1.133 | 1.162 | 1.189 | 1.112 | 1.215 |
| Sexual health clinic consultation | 1.000 | 1.000 | 1.000 | 1.000 | 1.000 | 1.000 | 1.000 | 1.000 | 1.000 | 1.000 |
| **5. Access to healthcare professional** | | | | | | | | | | |
| Phone | 0.949 | 0.937 | 0.962 | 1.001 | 0.902 | 0.946 | 0.942 | 0.926 | 0.947 | 0.953 |
| Instant messenger | 1.027 | 0.994 | 1.062 | 1.108 | 0.977 | 1.001 | 1.025 | 0.991 | 1.045 | 1.025 |
| Email | 1.048 | 1.034 | 1.062 | 1.078 | 1.038 | 1.030 | 1.069 | 1.001 | 1.049 | 1.054 |
| Face to face | 1.000 | 1.000 | 1.000 | 1.000 | 1.000 | 1.000 | 1.000 | 1.000 | 1.000 | 1.000 |

Continued

**Table 3** Continued

| Attribute and levels | Full data set | Male | Female | 16–18 | 19–21 | 22–24 | Single | One partner | Previous test | No previous test |
|---|---|---|---|---|---|---|---|---|---|---|
| | 1230 | 607 | 623 | 415 | 406 | 409 | 615 | 512 | 393 | 790 |
| | OR | OR | OR | OR | OR | OR | OR | OR | OR | OR |
| **6. How you get antibiotics** | | | | | | | | | | |
| Post to home | 1.011 | 0.996 | 1.028 | 0.977 | 1.010 | 1.048 | 0.957 | 1.055 | 1.011 | 1.020 |
| Post to collection point | 1.031 | 1.082 | 0.983 | 1.010 | 1.036 | 1.047 | 0.968 | 1.072 | 1.034 | 1.035 |
| Collect from pharmacy | 1.075 | 1.064 | 1.087 | 1.039 | 1.106 | 1.080 | 1.070 | 1.056 | 1.086 | 1.078 |
| Collect from sexual health clinic | 1.000 | 1.000 | 1.000 | 1.000 | 1.000 | 1.000 | 1.000 | 1.000 | 1.000 | 1.000 |

95% CIs provided in online supplementary file 3.
OR values not statistically significant are shaded blue.
For further information regarding the explanation of attributes and levels to young people in the questionnaire please see online supplementary file 2.
GP, general practitioner.

access antibiotics apart from a slight preference for the pharmacy versus sexual health clinic (OR 1.075, 95% CI 1.018 to 1.134).

## DISCUSSION

Our findings indicate that, based on the levels included in the DCE questionnaire, young people are willing to wait in order to have a chlamydia screening test result with a lower chance of a false negative result. The conclusion for new test developers is that time to result is less important than accuracy, and that test users are unlikely to prefer a point-of-care or self-screening test with lower accuracy than the tests currently available to them. There was a strong preference for remote access to testing, consultation and antibiotic prescriptions, although for accessing a health professional there was no preference between online and face-to-face methods. This suggests a remote online pathway is acceptable to young people, *as long as* test performance remains equivalent.

In the various hypothetical situations presented, respondents showed a preference for chlamydia self-testing, self-sampling and postal testing over attendance at a testing location. For accessing treatment, a general preference was exhibited for online versus traditional GP or pharmacy services over clinic services. For receipt of antibiotics, there was little difference in preferences. We were also able to identify which attributes people may be willing to trade to maximise their utility. Looking at the trade-off between accuracy and time to result, we found that young people are willing to wait noticeably longer in order to have a test result with a lower chance of a false negative result, reinforcing the need for test equivalence.

The strengths of this DCE study include, first, the robust methods employed to select attributes and levels, which aligned with the recommendations of Coast et al, ensuring attributes are 'manipulable in policy'.[38] Second, the fact that participants were drawn from the general population targeted by the NCSP, rather than from healthcare settings. This enabled us to access a demographically and geographically representative national sample of young people, including those who have had no previous contact with STI services. Finally, the use of an online panel enabled large-scale data to be collected at a reasonable cost and allowed validity checks that would not otherwise have been possible with postal questionnaire responses. The large sample size also allows comparison between several subgroups to explore differences based on age, gender, testing history and relationship category.

However, the study does have a few limitations. The first relates to the selection of attributes and levels. While the selection process employed was very rigorous, this cannot detract from the fact that further attributes were identified which might impact on individuals' choices. To mitigate against this, where such an attribute was excluded, information was provided in the survey background section to minimise respondents forming their own views on the impact of this attribute on the pathway.

**Table 4** Trade-off between accuracy and time (probability of uptake)

| | Full data set (%) | Male (%) | Female (%) | 16–18 (%) | 19–21 (%) | 22–24 (%) | Single (%) | One partner (%) | Previous test (%) | No previous test (%) |
|---|---|---|---|---|---|---|---|---|---|---|
| Pathways: (n in 100*/time†) | | | | | | | | | | |
| 5 in 100*/30 min† | 38 | 40 | 35 | 36 | 39 | 38 | 36 | 39 | 40 | 36 |
| 2 in 100/14 days | 58 | 55 | 62 | 59 | 58 | 59 | 59 | 58 | 58 | 59 |
| Not test | 4 | 5 | 3 | 5 | 3 | 3 | 4 | 3 | 2 | 5 |
| Total | 100 | 100 | 100 | 100 | 100 | 100 | 100 | 100 | 100 | 100 |

*Test accuracy=n in 100 people will be told they do not have chlamydia when they do.
†Time to receipt of result.

Use of an online panel also provided accurate records of the time taken to complete the survey, and permitted additional validity checks which would not otherwise have been possible with a written questionnaire. On the other hand, use of an online panel excluded young people who do not have access to the internet, thereby potentially over-representing the acceptability of online care. However, given the extremely high proportion (97%) of 15–24 year-olds accessing the internet daily via a mobile device[39] and owning a smartphone,[40] it is evident that the vast majority of the target population could access online care pathways if they choose to do so. The question of whether these young people have the degree of digital and health literacy needed for online testing and treatment was not explored in this study. Finally, information was unavailable on young people who opened but decided not to complete the questionnaire, so systematic comparison with those who responded was not possible.

Behavioural factors, such as embarrassment,[7] stigma[41] and privacy and anonymity concerns,[42 43] are known to influence uptake of sexual health services, as well as structural factors such as convenience and perceived barriers to access[8]; many of these may be lower for a non-face-to-face service. Balanced against this, access to online testing and treatment is valued less strongly than attributes such as test accuracy and time to result. In individual cases, a young person's preference for remote versus face-to-face testing and treatment might differ.

Finally, it is difficult to compare the present results with other published DCEs because of key differences in perspective. Miners et al (sample n=3358)[20] and Llewellyn et al (sample n=233)[21] both only focused on preferences within existing traditional service delivery models, and did not incorporate any hypothetical future scenarios (eg, point-of-care testing, self-testing or treatment via eHealth/mHealth solutions). Two other DCE studies, which did consider self-sampling at home for chlamydia screening,[18 19] with sample sizes 174 and 126, respectively, both described sending the sample to a laboratory for analysis rather than a self-test. One of these studies[19] did identify a stronger preference for attendance at a family planning clinic, rather than self-sampling, but since participants were recruited from the waiting room

of such a clinic they cannot be considered representative of the general population targeted by the NCSP. A fifth study, which examined preferences for point-of-care testing, only surveyed clinicians undertaking STI testing (sample n=218), thus excluding the population actually targeted for chlamydia screening.[44] The only stated preference study that has considered patients' preferences for STI self-testing focused on HIV (sample n=365).[45] The authors reported results in line with our findings, with respondents exhibiting a preference for tests which are accurate, timely and private/anonymous. However, this study was undertaken in 2002 prior to smartphones and at a time when self-testing for HIV was still under development. Importantly for screening tests, only one DCE study to date[21] has sought to include non-service users (ie, populations with no experience of STI testing). The authors identified a preference for testing for all STIs, in settings with healthcare professionals with specialist knowledge present, and for receipt of negative as well as positive results. This study used a convenience sample of 233 students from two universities which was unlikely to be geographically or socioeconomically representative. All other studies have drawn their DCE samples from people who were either current service users or attendees for other linked services. Use of such samples represents a significant shortcoming when considering the introduction of new technologies for asymptomatic chlamydia testing and management. Most sample sizes were also smaller than in the present study.

A number of questions remain which our research does not address. First, recognising that the range of STIs included is an important consideration for young people in choosing to test,[20 21] further research is required to understand this better in the context of potential new screening pathways which incorporate other STIs (eg, gonorrhoea). Second, our study highlights a number of methodological considerations where there is an absence of consensus that may warrant further exploration to improve consistency. These include the number of choice sets to include in a DCE and the use of repeated choice sets as an internal validity measure. Finally, given that cost was not included as an attribute in this study, it is not possible to provide an indication of willingness to pay, or

the monetary benefits of potential service changes, from this DCE.[46]

The present study is, to our knowledge, the first to present a large-scale, quantitative analysis of young people's preferences for attributes of potential new pathways to deliver testing and treatment of asymptomatic chlamydia, based on a nationally representative population. The DCE methodology applied also produced a measure of the relative strength of preference between different attributes and levels, and potential trade-offs. This can provide useful evidence to technology developers, policymakers, commissioners and service providers. In particular, it provides a first insight into preferences for the type of technologies currently under development, and those which might be available for use in the near future, compared with the features of existing products and services. This can indicate how young people may respond to changes in pathways and to the introduction of new technologies.

Within the context of current UK sexual health policy and commissioning of sexual health services,[12] this DCE provides supportive evidence for the policy direction of remote chlamydia testing and treatment. However, while young people overall expressed a stronger preference for attributes such as self-test, online consultation, etc, a small proportion still preferred existing pathways. This suggests that, in order to maximise benefit, face-to-face services should continue to be available in addition to any online screening and treatment service. This will ensure that services are inclusive and accessible irrespective of digital/health literacy, while recognising that people's needs and preferences may change depending on their personal circumstances.

**Author affiliations**
<sup>1</sup>Warwick Medical School, University of Warwick, Coventry, UK
<sup>2</sup>Scientific Advice, National Institute for Health and Care Excellence, London, UK
<sup>3</sup>Research Department of Infection and Population Health, University College London, London, UK
<sup>4</sup>Department of Psychology, Glasgow Caledonian University, Glasgow, UK
<sup>5</sup>Applied Diagnostic Research and Evaluation Unit, Institute for Infection and Immunity, St George's University of London, London, UK
<sup>6</sup>Enterprise and Innovation Group, Coventry University, Coventry, UK

**Acknowledgements** We thank Dr Joshua Pink, National Institute for Health and Care Excellence (NICE), Manchester, UK for providing specialist advice on DCE design and analysis. We also thank academics in the Studies in Adolescent Sexual Health (SASH) Group at Coventry University who provided advice on how to access the young people who participated in focus groups; and YouthSight for their expertise in converting the questionnaire into an online format. Finally, the authors thank all the young people who took the time to participate in the focus groups and cognitive testing, and complete a detailed questionnaire describing their preferences, and the experts who contributed their views in the expert groups.

**Contributors** SE conceived and undertook the study as part of her PhD research. AS was the lead supervisor and takes responsibility for the integrity of the data and the accuracy of the data analysis. SP, LO and AS oversaw the DCE design and analysis. DB guided the qualitative research. CSE and JG provided clinical input into the development of the DCE materials and assisted with interpretation of findings. SE, AS, SP, LO and DB wrote the first draft of the article. STS (principal investigator), AS and CSE were applicants on the eSTI2 Consortium grant, wrote the initial protocol and obtained funding. All authors contributed to and agreed with the final version of the manuscript.

**Funding** The Electronic Self-Testing Instruments for Sexually Transmitted Infection Control (eSTI2) Consortium (which funded SE's studentship and also AS, JG, CSE and STS) was funded under the UKCRC Translational Infection Research (TIR) Initiative supported by the Medical Research Council (grant number G0901608) with contributions to the Grant from the Biotechnology and Biological Sciences Research Council, the National Institute for Health Research on behalf of the Department of Health, the Chief Scientist Office of the Scottish Government Health Directorates and the Wellcome Trust.

**Disclaimer** The funding body had no role in the design of the study, analyses, interpretation or decision to submit the manuscript for publication.

**Competing interests** None declared.

**Patient consent** Not required.

**Ethics approval** Biomedical and Scientific Research Ethics Committee at the University of Warwick (Ref: REGO-2015-1647).

**Provenance and peer review** Not commissioned; externally peer reviewed.

**Data sharing statement** Requests for access to data should be addressed to the corresponding author.

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
