## [Reviewer comments · BMJ Open]

This paper was submitted to a another journal from BMJ but declined for publication following peer review. The authors addressed the reviewers' comments and submitted the revised paper to BMJ Open. The paper was subsequently accepted for publication at BMJ Open.

(This paper received three reviews from its previous journal but only two reviewers agreed to published their review.)

ARTICLE DETAILS

TITLE (PROVISIONAL)	Young people's preferences for the use of emerging technologies for asymptomatic regular chlamydia testing and management: A discrete choice experiment in England
AUTHORS	Eaton, Sue; Biggerstaff, Deborah; Petrou, Stavros; Osipenko, Leeza; Gibbs, Jo; Estcourt, Claudia; Sadiq, S. Tariq; Szczepura, Ala

VERSION 1 – REVIEW

REVIEWER	Gill ten Hoor Maastricht University, The Netherlands
REVIEW RETURNED	06-Jun-2018

GENERAL COMMENTS	Thank you for allowing me to review this interesting paper. In this paper, the authors use a discrete choice experiment to examine chlamydia screening and treatment preferences in young people. Although this paper has some strong points, I do believe that there are aspects that needs to be improved: 1. The authors failed to include the available research about screening behavior and behavioral determinants. Earlier studies have shown that testing behavior is reasoned behavior: those who have a high testing intention, will subsequently test; those who have a high testing intentions, have good reasons to test. Searching for the best treatment preferences for this specific group (i.e. people at high risk) seems useful – a better rationale is needed for why preferences of the general population are examined.2. Additionally to point 1: people only have to test for chlamydia if they participate in high-risk behavior. Additionally, earlier research suggests that a general population approach will not be cost effective. – In this study, a general population was approached to figure out the preferences of a very specific group of people. The current methodology does not allow the authors to draws conclusions about the specific group that needs to test.3. The authors describe that they have done two literature reviews, a focus group, expert reviews and a narrative synthesis, these studies are not described/reported and can therefore not be evaluated. With that, it is also impossible to evaluate whether the right methodology was used in this study.4. The authors have to explain the following statement more elaborately: "A full factorial design was ruled out in favour of a fractional factorial design because a full factorial design would
---

	have contained 3,072 possible alternatives, which would have been unmanageable in practice for individuals to complete or for a blocked questionnaire format to handle. The smallest 100% efficient design that could be created included 48 choice sets. “ ◇ how did the authors end up with 48 sets? 5. More information is needed about the sample size calculation. 6. 490 people commenced the questionnaire, but did not complete it. The authors have to explain how they handled incomplete questionnaires
--	---

REVIEWER	A Miners LSHTM, UK
REVIEW RETURNED	20-Jul-2018

GENERAL COMMENTS	This is an interesting DCE. The design process is detailed however I have a number of concerns regarding the logic / meaning of the chosen attributes which means I am left questioning its usefulness. My comments are written in no particular order I can see why a DCE could be helpful, but what evidence is there that responses are any other than hypothetical? When is a remote test medically appropriate / how was the DCE question framed? Did removal of implausible designs only affect the efficiency? It seems strange to me that they can be removed (how many were there) without impacting the ability of the data to be analysed? Particularly when the design is looked at lots of choice sets would have to have been removed? More detailed is required as to how the sample were recruited. That is, by being part of the panel they are by definition unlikely to be representative of the general population ? Were different versions of the questionnaire (ie with different question / attribute ordering) used? This is where I get a little confused. Its fine to use dummy coding, but the limit with it is that the alternative specific constant (here I'm assuming it reflects the preference for either testing at all or not testing) can not be identified since it is confounded with the existing base levels. I'm also very surprised since no testing was included as a label, why no comment was made on how the various attribute levels impact upon uptake? Particularly since this rationale was given in the introduction. Now I have read on I see it is in table 4 although there is no mention of how it was calculated, or its meaning. Surely the emphasis should be on the change in uptake given different attribute levels? ORs should be quoted with Cis and where ORs are written, the reference level should be made clear eg x was preferred to why, rather than x alone. P values and CI should be included in the atables. Were other model forms considered (other than the MNL) given that the some of the options eg A and B are more closely related than not testing eg. The nested logit?
---

	There is almost no description of the chosen attributes and levels in the methods section. What is meant by self-sample and work and IM? Some of the attributes and levels seem to be inherently linked, and I'm unclear how this was dealt with in the analysis. For example, if someone attends a pharmacy for a treatment consultation then surely the Abs will be picked up from there? I see from the example that this wasn't the case. Personally I don't think this makes any sense at all. Similarly I would have thought someone attending a GPs will also pick them up from a pharmacy. Further, I don't understand the difference between some of the attributes, which goes back to my question about the attributes and levels being poorly described. For example, if a person has been to a GPs to give a sample what role does the pharmacy have or need when it comes to treatment other than giving the Abs? Why take a sample to a pharmacy but go to a SH for a treatment consultation (unless the two are together)?
--	--

VERSION 1 – AUTHOR RESPONSE

REVIEWERS' FEEDBACK		
Ref	Reviewer comments	Author response and changes made
R1	Reviewer: 1 Gill ten Hoor; Institution and Country: Maastricht University, The Netherlands	
	Although this paper has some strong points, I do believe that there are aspects that needs to be improved	
R1.1	(i) The authors failed to include the available research about screening behavior and behavioral determinants. Earlier studies have shown that testing behavior is reasoned behavior: those who have a high testing intention, will subsequently test; those who have a high testing intentions, have good reasons to test. (ii) Searching for the best treatment preferences for this specific group (i.e. people at high risk) seems useful – a better rationale is needed for why preferences of the general population are examined	(i) We agree with the reviewer that screening behaviour is a complex issue. Determinants governing this are linked to both individuals' characteristics and also, as explored in the present DCE, service characteristics. We are aware of the likely influence of psychological mediators and we do reference this (see lines 15-18 and lines 318-323 in the Discussion section). (ii) The rationale for our focus on the preferences of the general population should be evident from the Introduction & is now stated more clearly in the study aim (line 56). Our DCE focuses on preferences in relation to the UK National Chlamydia Screening Programme, rolled out 2003–08, which targets the general population of young people up to the age of 24 years. However, as in the Netherlands where a national chlamydia screening programme was launched in 2008, participation rates have been falling and a low level of uptake raises questions over whether national screening, as currently delivered, is cost-effective.

		However, with new digital and mHealth technologies on the horizon, there is an opportunity to change the ways in which screening & treatment are delivered (e.g. selftesting combined with an online care pathway). To date, it is unclear what impact such reengineering might have on future uptakes. Information on the preferences of UK young people invited for screening in terms of these potential future changes is therefore important to consider, prior to introduction. The present DCE study examines this. Our DCE was designed to quantify the preferences of a representative sample of those currently targeted for chlamydia screening i.e. in terms of age, gender, ethnicity, geographical location. Interestingly, in England evidence of a reduction in chlamydia prevalence concurrent with large-scale population testing has been reported since our original submission; additional text (lines 10-11) and new reference 6 have been added.
--	--	---

R1.2	Additionally to point 1: people only have to test for chlamydia if they participate in high-risk behavior. Additionally, earlier research suggests that a general population approach will not be cost effective. – In this study, a general population was approached to figure out the preferences of a very specific group of people. The current methodology does not allow the authors to draw conclusions about the specific group that needs to test.	We would not agree that only those who participate in 'high-risk behaviour' need to be tested for chlamydia. Therefore, our current methodology does allow us to draw conclusions about the "specific group that need to test". As pointed out in R1.1 above, our DCE study is set within the context of a national chlamydia screening programme and the potential introduction of disruptive technologies (i.e. self-testing and an online care pathway). The 2007 HTA study of risk factors for chlamydia screening "did not find any factors, other than young age, that would help to target screening more easily" (Low et al 2007). Although we are interested in cost-effectiveness, this is not the focus of the present paper. We did not include a 'cost' attribute in our DCE to enable us to assess willingness-to-pay. This was partly because this is recognised as difficult to include in countries like the UK, where healthcare is free at the point of delivery, and also inclusion of this attribute in DCEs has fallen in recent years because of such issues (ref 35). However, we are well aware of ongoing discussions about the cost-effectiveness of general population screening for chlamydia as currently delivered, and how future service changes might influence this. For example, although rapid point-of-care tests are not reported to be more cost-effective than the
------	---	--

		status quo, economic modelling indicates they may become so if they can increase uptake rates (Hislop et al 2010). We will shortly be reporting the results of our own economic modelling of a fully remote online pathway for testing & treatment of chlamydia. Nevertheless, we agree with this reviewer that modifying behaviour in the high-risk group (i.e. those who test positive after general population screening for chlamydia) is important in terms of reinfection. Two authors[AS, TS] are therefore currently involved in the development of an 'add-on' online mHealth intervention to increase condom use among young people using online chlamydia self-testing services (Crutzen et al 2018).
R1.3	The authors describe that they have done two literature reviews, a focus group, expert reviews and a narrative synthesis, these studies are not described/reported and can therefore not be evaluated. With that, it is also impossible to evaluate whether the right methodology was used in this study.	Due to the word limit we were only able to provide summary details of these elements in the manuscript (lines 85-115). However, this reviewer may be interested to know that an article is in progress fully describing the methodologies more fully, which will be peerreviewed. A description of both literature reviews is provided in the current article. The first adopted a systematic review approach & the second was a scoping review; the protocol for the former has been published (PROSPERO Reg: CRD42014014862). We consider that the focus groups are sufficiently described, within the word limit (lines 92-98). We now make it clear that there were four focus groups, not one (line 93).
		The role of the expert groups & the overview of methods used for narrative synthesis are, we consider, adequately described (lines 100-115).
R1.4	The authors have to explain the following statement more elaborately: "A full factorial design was ruled out in favour of a fractional factorial design because a full factorial design would have contained 3,072 possible alternatives, which would have been unmanageable in practice for individuals to complete or for a blocked questionnaire format to handle. The smallest 100% efficient design that could be created included 48 choice sets." □ how did the authors end up with 48 sets?	We appreciate that this reviewer, along with some other readers, may not be familiar with DCEs. We have therefore added a June 2018 review of the impact of design characteristics on statistical efficiency for interested readers (new ref 31, Vanniyasingam et al 2018). To explain questionnaire design: In Discrete Choice Experiments it is usually not feasible to present the respondent with choice sets which include all possible combinations since the number would be unmanageably large. Therefore, accepted practice when designing the questionnaire is to identify the minimum number of choice sets which can provide an efficient design (48 in this case). Recognising that 48 choice sets is likely to be too many for participants to complete, the

		questionnaire had to be blocked (divided up into smaller questionnaires). 48 can be blocked into two questionnaires with 24 sets, three questionnaires of 16 sets etc. Cognitive testing was undertaken based on two questionnaires of 24 sets. This confirmed that a study based two such questionnaires was acceptable to participants so this was used. We have inserted extra text to clarify this (lines 137-139).
R1.5	More information is needed about the sample size calculation.	DCE sample size calculations are complex and, unlike randomised controlled trials, there is no specified method by which they should be calculated (ref 27), although there is guidance on sub-group size. In the present study, two demographic characteristics were identified a priori in the sampling frame for sub-group analysis – gender and age range. Where sub-group analysis is planned, it is recommended that there should be a minimum of 200 respondents in each sub-group (ref 33). Therefore, to reach a minimum of 200 respondents for each of the gender (M/F) and agespecific sub-groups (16-18; 19-21; 22-24 years) a total sample size of 1,200 was required, 600 per questionnaire. We have inserted extra text to clarify this (lines 170-171).
R1.6	490 people commenced the questionnaire, but did not complete it. The authors have to explain how they handled incomplete questionnaires	In total 1,230 completed responses were achieved by YouthSight from their panel of young people within the target gender/age range groups. The platform analytics showed that 460 people had opened the questionnaire but did not complete the survey. No further information was available on the demographics of these participants nor any information on the point at which they chose to exit the survey. Therefore, these people could not be included in our analyses. We have inserted extra text to clarify this (lines 210-212).

R2	Reviewer: 2 A Miners Institution and Country: LSHTM, UK	
	This is an interesting DCE. The design process is detailed however I have a number of concerns regarding the logic / meaning of the chosen attributes which means I am left questioning its usefulness. My comments are written in no particular order	We thank the reviewer for these comments.

R2.1	I can see why a DCE could be helpful, but what evidence is there that responses are any other than hypothetical? When is a remote test medically appropriate / how was the DCE question framed?	Responses will inevitably be 'hypothetical' since the DCE is exploring preferences for emerging technologies & digital service which are not currently available. This is now clearly stated (see line 52) Even for current services, responses would be 'hypothetical' for the majority since 64% of respondents reported they had no previous experience of being tested (see Supplementary File 2, part 1). In terms of framing the DCE, our study was undertaken as part of a larger programme funded under the UKCRC Translational Infection Research Initiative to advance development of a remote self-test & online care pathway for chlamydia (see Funding, page 12). Appropriate remote self-testing & online care might improve screening uptake & treatment completion, and also potentially reduce chlamydia screening and treatment costs. A remote online clinical pathway has been demonstrated as medically appropriate by our research team (ref 14) & development of a remote chlamydia self-test is underway.
R2.2	Did removal of implausible designs only affect the efficiency? It seems strange to me that they can be removed (how many were there) without impacting the ability of the data to be analysed? Particularly when the design is looked at lots of choice sets would have to have been removed?	In terms of implausible combinations, expert review identified only four:  • Attribute 1 (Self-Test), with Attribute 2 (7 Days) • Attribute 1 (Self-Test) with Attribute 2 (14 Days) • Attribute 1 (Self-Sample & Post for Analysis) with Attribute 2 (30 mins) □ Attribute 1 (Self-Sample & Post for Analysis) with Attribute 2 (2 hours). In addition, a number of unlikely combinations were also identified, for example:  • Consultation Method – Pharmacy • Access to a HCP – Face to Face • How you get your antibiotics – collect from sexual health clinic. Whilst the example above is feasible, it is unlikely that a person attending a pharmacy for a consultation would then go to a second location to get their antibiotics when they could be dispensed by the pharmacist. As explained (lines 141-144), both implausible and unlikely combinations were included in cognitive testing in the pilot phase. It was identified that implausible combinations did
		impact on completion of the task (so these were removed), but that unlikely combinations did not impact on completion (so these were not removed).

R2.3	More detailed is required as to how the sample were recruited. That is, by being part of the panel they are by definition unlikely to be representative of the general population?	This is an important point to address. An online research panel was chosen as the best route to recruit participants in a timely manner, recognising the large sample size required, and the need to access a representative sample of the general population across the UK. This DCE was designed to explore the preferences of the general population targeted by the national chlamydia screening programme, including young people who have not tested to date. Supplementary Information File 2 (Part 1) compares the characteristics of the DCE sample and those of 16-24 year olds in England. This indicates that gender, age and ethnicity breakdown is broadly in line with national population demographics, with a slight underrepresentation of the 16-18 year age group. In terms of geographical location, the sample is also broadly in line with the geographical distribution of 16-24 year olds identified in the ONS Mid-Year Population Estimates 2015, with the exception of a lower proportion of respondents from the East of England. A similar, nationally representative sample could not easily be achieved through recruitment via healthcare or other settings. Also, online recruitment allowed sampling quotas to be set; data collection was paused after 50 responses on each questionnaire to test the model was meeting sampling quotas. Although it may it may be argued that the use of an online panel limits generalisability by excluding populations with no access to the internet, 97% of 15-24 year-olds in England currently access the internet daily via a mobile device and own a smartphone (lines 312313). So, use of online panel will exclude very few of the target population for screening. In contrast, all other DCE studies have drawn their samples from current STI service users or those attending other linked services. The preferences of such populations will not be representative of those targeted for chlamydia screening. However, one third of our DCE sample (32%) report previous experience of being tested for STIs (see Supplementary File 2, part 1). National figures indicate that ~25% of the population aged 20-24 were in contact with local reproductive health services in 2015/16 (Hagell et al 2017).
R2.4	Were different versions of the questionnaire (ie with different question / attribute ordering) used? This is where I get a little confused. Its fine to use dummy coding, but the limit with it is that the alternative specific constant (here I'm assuming it reflects the preference for either testing at all or	To avoid confusion, we now draw attention to Table 4 at the outset (along with Table 3) so that other readers do not experience the same problem (line 229-230). As explained in R1-4 and lines 128-134 in the manuscript, the 48 choice sets were blocked into two questionnaires with 24 sets each. Outputs for optimum questionnaire design identified in SAS 9.4 were used to compute the design in SAS JMP Pro 11.2.0. Thirty

	not testing) can not be identified since it is confounded with the existing base levels. I'm also very surprised since no testing was included as a label, why no comment was made on how the various attribute levels impact upon uptake? Particularly since this rationale was given in the introduction. Now I have read on I see it is in table 4 although there is no mention of how it was calculated, or its meaning. Surely the emphasis should be on the change in uptake given different attribute levels?	questionnaire design runs were completed and the design with the highest number of levels balanced across both questionnaire blocks was selected manually. Choice set 1 was repeated as choice set 25 in each questionnaire for validity checks. Not included (due to word limits) was which repeat choice set (1 or 25) was dropped from final analysis. In the absence of a clearly established method, choice set 1 was dropped (assumed at start of questionnaire participants are less familiar with survey approach). However, both were compared. Only one statistical significant difference was found: for how you get antibiotics, 'post to collection point' was not significant for choice set 25 (OR 1.031, 95% CI 0.981-1.085) but was for choice set 1 (OR 1.053, 95% CI 1.002-1.106). Finally, although we did not examine how different attribute levels might change uptake, we did examine trade-offs between accuracy and time to result since these are the most important attributes likely to influence the decision to test, and participants expressed the strongest preference for these (lines 234-240). We examined uptake probability for tests with characteristics at the opposite ends of the spectrum and report "willing to wait noticeably longer in order to have a test result with a lower chance of a false negative result" (line 242-244).
R2.5	ORs should be quoted with CIs and where ORs are written, the reference level should be made clear eg x was preferred to why, rather than x alone. P values and CI should be included in the tables.	Apologies. Although we include 95% CI values in the abstract, in the body of the main manuscript text we only referred the reader to Supplementary file 2: "Additional DCE results, including the 95% CIs for all sub-group analysis are presented in Supplementary Information File 2" We have now added 95% CI values to all OR values quoted in the main text, and also specify the referent (line 234-266). We also make it clear under the Table 3 heading that 95% CI for OR values in this table are presented in Supplementary file 2 (Part 3).
R2.6	Were other model forms considered (other than the MNL) given that the some of the options eg A and B are more closely related than not testing eg. The nested logit?	As described in the Statistical Analyses section (line 177), we selected the MNL model because this is the convention for choice sets including three or more alternatives ('option A', 'option B' or 'I would not test' as in our case). This has been identified as the core choice model used for the analysis of stated preference data in a variety of fields (ref 34, and its use in healthcare has doubled in the period 2001-2012 rising to 44% published studies (ref 35). Sadly, it was beyond the scope of this research to compare a range of other models, although we recognise that, particularly because of the 'I would not test option', there are other modelling approaches that it may be useful to explore. The nested logit model which groups "subsets of alternatives that are more similar to each other with respect to excluded

		characteristics than they are to other alternatives” (ref 26) is one that would warrant further investigation.
R2.7	There is almost no description of the chosen attributes and levels in the methods section. What is meant by self-sample and work and IM?	Apologies. Due to the article word limit it was not possible to provide a description of the attributes and levels in the main manuscript. Full descriptions are provided in Supplementary Information File 1, but this was perhaps not clear. We now have a separate section ‘Final Attributes and Levels Selected’ and signpost the Supplementary Information File more clearly (lines 160-163). We also make it clear under the Table 1 heading that full descriptions are provided in Supplementary Information File 1 (pages 3-7).
R2.8	Some of the attributes and levels seem to be inherently linked, and I’m unclear how this was dealt with in the analysis. For example, if someone attends a pharmacy for a treatment consultation then surely the Abs will be picked up from there? I see from the example that this wasn’t the case. Personally I don’t think this makes any sense at all. Similarly I would have thought someone attending a GPs will also pick them up from a pharmacy. Further, I don’t understand the difference between some of the attributes, which goes back to my question about the attributes and levels being poorly described. For example, if a person has been to a GPs to give a sample what role does the pharmacy have or need when it comes to treatment other than giving the Abs? Why take a sample to a pharmacy but go to a SH for a treatment consultation (unless the two are together)?	This question is addressed more fully in R2.2 above: Although the examples mentioned here are unlikely , they are feasible i.e. that a person attending a pharmacy for a treatment consultation might then go to a second location to get their antibiotics even though they could be dispensed by the first pharmacist. Cognitive testing examined the impact of unlikely combinations like this on completion (see R2.2). This testing identified that, although implausible combinations did impact on completion of the task (therefore these were removed), unlikely combinations did not (so these remained).

ADDITIONAL REFERENCES

- Crutzen, R., Newby, K.N., Brown, K., Bailey, J., Hunt, J., Alston, T., Saunders, J., Sadiq, T., Das, S., & Szczepura, A. Wrapped: Development of an intervention to increase condom use amongst users of online chlamydia self-testing services. Oral presentation at the *8th Conference on Supporting Health by Technology*, Enschede, the Netherlands. 1 June 2018.
- Hagell A, Shah R and Coleman J (2017) *Key Data on Young People 2017*. London: Association for Young People’s Health. Access at: <http://www.ayph.org.uk/keydata2017/FullVersion2017.pdf>
- Hislop J, Quayyum Z, Flett G, Boachie C, Fraser C, Mowatt G. Systematic review of the clinical effectiveness and cost-effectiveness of rapid point-of-care tests for the detection of genital Chlamydia infection in women and men. *Health Technol Assess* 2010; 14(29).
- Low N, McCarthy A, Macleod J, Salisbury C, Campbell R. Epidemiological, social, diagnostic, and economic evaluation of population screening for genital chlamydial infection. *Health Technol Assess* 2007; 11(8)

VERSION 2 – REVIEW

REVIEWER	Gill ten Hoer Maastricht University, The Netherlands
REVIEW RETURNED	09-Sep-2018

GENERAL COMMENTS	Although the authors did improve their article, I am still not satisfied with many of the author's responses on my feedback. 1.1. (i) By just adding the word 'behavioral' in sentence 15-18, the authors did not include available research about chlamydia screening behavior and its determinants. (ii) The authors wrote "The aim of the present study was to undertake a comprehensive assessment of the preferences of young people, targeted by the National Chlamydia Screening Programme, for emerging technology options for testing and treatment in the context of a "check-up test" where remote care could be medically appropriate.¹⁴". With this statement, the authors should still make clear why their research focuses on a large group of people (people at low or no risk) instead of the risk population. What is then the additional value of this research? 1.2. The authors disagree with my point of feedback. That is OK, but (in my opinion) they should at least describe our disagreement in their discussion. 1.3. The authors explain that they only give a summary of their earlier research because of the word limit. With that, it is still impossible to evaluate whether the right methodology was used in the current study. 1.4. About the 48 sets. My apologies for not being clear about my question. I do understand that it was impossible to include all possible combinations, but more information is needed about the statement "The smallest 100% efficient design that could be created included 48 choice sets." 1.5. This point of feedback has been answered sufficiently. 1.6. About the missing data. Thank you for this clarification. However, because this data is not available, it is unclear whether there was a systematic missingness. This should be added to the limitations.
--

REVIEWER	Alec Miners LSHTM, UK
REVIEW RETURNED	12-Sep-2018

GENERAL COMMENTS	The authors have done a reasonable job of answering the questions. However, if were to be picky I would say its a shame that they didn't undertake further analysis as I had previously indicated. That is, to look at the impact of the individual test characteristics on uptake, and to have undertaken further analysis not just the MNL. The MNL in this context should be the minimum.
---

VERSION 2 – AUTHOR RESPONSE

REVIEWERS' FEEDBACK		
Ref	Reviewer comments	Author response and changes made

R1	Reviewer: 1 Gill ten Hoor, Maastricht University, The Netherlands	
R1.1	(i) By just adding the word 'behavioral' in sentence 15-18, the authors did not include available research about chlamydia screening behavior and its determinants. (ii) The authors wrote "The aim of the present study was to undertake a comprehensive assessment of the preferences of young people. targeted by the National Chlamydia Screening Programme, for emerging technology options for testing and treatment in the context of a "check-up test" where remote care could be medically appropriate.¹⁴". With this statement, the authors should still make clear why their research focuses on a large group of people (people at low or no risk) instead of the risk population. What is then the additional value of this research?	(i) In terms of research on screening behaviour, the manuscript already draws the reader to research on behavioural determinants (see Introduction and Discussion). Since this is a quantitative DCE study of preferences, we do not consider that a more in depth discussion of psychological determinants is required. Reviewer 2 did not highlight this as an issue. (ii) As mentioned previously, our DCE study is set within the context of the UK national chlamydia screening programme (NCSP). The population screened is all young people up to the age of 24 years. It should therefore be evident why our study includes 'people at low or no risk'. The at risk population targeted by the NCSP includes all young people, so will inevitably include some 'people at low or no risk'. Our DCE successfully recruited a very large sample of this population i.e. 1,230 young people aged 16-24 (representative in terms of age, gender, ethnicity and geographical location). If the reviewer is questioning a national age-based chlamydia screening policy, we consider discussion of this is outside the remit of this DCE study.
R1.2	The authors disagree with my point of feedback. That is OK, but (in my opinion) they should at least describe our disagreement in their discussion.	The reviewer's previous point was that: "Additionally to point 1: people only have to test for chlamydia if they participate in highrisk behavior. Additionally, earlier research suggests that a general population approach will not be cost effective. In this study, a general population was approached to figure out the preferences of a very specific group of people. The current methodology does not allow the authors to draws conclusions about the specific group that needs to test." As indicated in our response R1.1 (ii) above, population-based screening does not require that 'people only have to test for chlamydia if they participate in high-risk behavior'. Therefore, our conclusions are relevant to 'the specific group that needs to test'.
R1.3	The authors explain that they only give a summary of their earlier research because of the word limit. With that, it is still impossible to evaluate whether the right methodology was used in the current study.	We assume that this refers to the methodologies used in the pre-DCE stages to select attributes and levels: (i) secondary research i.e. systematic/scoping literature reviews; (ii) primary qualitative research i.e. focus groups/expert groups/narrative synthesis.

		We appreciate that a unique aspect of our DCE study is the very thorough earlier stages undertaken. Therefore, although stage both are fully referenced, we now provide additional information in Supplementary file 1 (Selection of Attributes and Levels).
R1.4	About the 48 sets. My apologies for not being clear about my question. I do understand that it was impossible to include all possible combinations, but more information is needed about the statement "The smallest 100% efficient design that could be created included 48 choice sets."	Methods for identifying the most efficient combination are a standard element of DCE design. For any readers who are not familiar with this aspect of DCE design, we had previously added a reference reviewing these (Vanniyasingam et al 2018). We do not consider further explanation is required in the manuscript. Reviewer 2 did not highlight this as an issue.
R1.5	This point of feedback has been answered sufficiently.	Thank you.
R1.6	About the missing data. Thank you for this clarification. However, because this data is not available, it is unclear whether there was a systematic missingness. This should be added to the limitations.	This is now mentioned as a limitation to the use of recruitment via online panels (lines 317319).
R2	Reviewer: 2 A Miners, London School of Hygiene & Tropical Medicine, London, UK	
R2.1	No further comment	Thank you.
R2.2	No further comment	Thank you.
R2.3	No further comment	Thank you.
R2.4	The authors have done a reasonable job of answering the questions. However, if were to be picky I would say its a shame that they didn't undertake further analysis as I had previously indicated. That is, to look at the impact of the individual test characteristics on uptake, and to have undertaken further analysis not just the MNL. The MNL in this context should be the minimum	Although we agree that it would have been interesting to undertake such further analyses, we did not have the capacity to do this.
R2.5	No further comment	Thank you.
R2.6	No further comment	Thank you.
R2.7	No further comment	Thank you.
R2.8	No further comment	Thank you.